# Knowledge, Attitudes, and Practices of High-Risk Patients towards Prevention and Early Detection of Chronic Kidney Disease (CKD) in Saudi Arabia

**DOI:** 10.3390/ijerph20010871

**Published:** 2023-01-03

**Authors:** Abdullah Alghamdi, Abdullah Alaryni, Khalid AlMatham, Osamah Hakami, Rayan Qutob, Abdullah Bukhari, Amani Abualnaja, Yara Aldosari, Noora Altamimi, Khawlah Alshahrani, Areej Alsabty, Amal Abdullah

**Affiliations:** 1Medical College, Imam Mohammad Ibn Saud Islamic University, Riyadh 11564, Saudi Arabia; 2King Fahad Medical City, Alfaisal University, Riyadh 11533, Saudi Arabia; 3Security Forces Hospital, Riyadh 11481, Saudi Arabia

**Keywords:** attitude, chronic kidney disease, knowledge, practice, CKD

## Abstract

Context: Chronic kidney disease (CKD) is characterized by the presence of kidney damage or decreased kidney function. In the Kingdom of Saudi Arabia, the prevalence of CKD is at 5.7%, which represents a high burden on health care systems. Aims: This study aimed to assess the knowledge, attitudes, and practices of high-risk patients towards prevention and early detection of chronic kidney disease in Saudi Arabia. Setting and Design: Descriptive cross-sectional study in Saudi Arabia. Methods and Material: This study was designed using a newly developed instrument, the CKD Screening Index. It was conducted from December 2021 to May 2022 by a self-administered questionnaire. The questionnaire has three parts: socio-demographic data, clinical factors, and the CKD screening index tool. Statistical analyses used: Independent *t*-test, One-Way ANOVA, LSD, Games–Howell tests. Results: Knowledge of kidney function had a significant difference across patient groups with varying employment status. Monthly income is a significant factor for the patient attitude on healthcare towards preventing kidney disease. On the other hand, educational level significantly affects the overall attitude of patients towards preventing kidney disease. Conclusion: Understanding knowledge, attitudes, and practices associated with CKD is vital to informing optimal policy and public health responses in the country.

## 1. Introduction

Chronic kidney disease (CKD) is defined as the presence of kidney damage or decreased kidney function (defined as estimated glomerular filtration rate (eGFR) of less than 60 mL/min per 1.73 m^2^) for 3 or more months, irrespective of the cause. Kidney damage refers to pathologic abnormalities, whether established via kidney biopsy or imaging studies, or inferred from markers such as urinary sediment abnormalities or increased rates of urinary albumin excretion. Globally, the prevalence of CKD is 9.1%, and there were 697.5 million cases of CKD (all stages) reported worldwide [1], while in the Kingdom of Saudi Arabia, the prevalence of CKD is 5.7%, according to an epidemiological study done in 2010 [2], which represents a high burden on health care systems. Over the last few years, it has become well established in the medical literature and community that CKD is related to an increased risk of premature mortality [3]. Because the majority of CKD patients are not clinically identified, this global growth in the prevalence of CKD necessitates improvements in the global approach to CKD prevention, primarily via recognizing risk factors, because the majority of CKD cases were not clinically recognized, mainly because of the lack of patients’ awareness about CKD risk factors. The most important risk factors for developing CKD are hypertension and diabetes mellitus. These two diseases promote vascular alterations that increase the risk for macro and micro vascular complications including kidney impairment.

However, there is limited information on the knowledge, attitudes, and practices towards prevention and early detection of CKD in Saudi Arabia. As such, this study aimed to assess the knowledge, attitudes, and practices of high-risk patients towards prevention and early detection of chronic kidney disease in Saudi Arabia and to find out the association between the participants’ knowledge and attitude with their socio-demographic characteristics.

## 2. Materials and Methods

This descriptive cross-sectional study was designed to assess knowledge, attitudes, and practices perceived by Saudi patients at risk for CKD using a newly developed instrument, the CKD Screening Index, which was validated, and its reliability was checked in previous studies. In the previous studies, its internal consistency was 0.87 and 0.8 in Palestine [4] and Jordan [5]. There were three parts to the questionnaire: socio-demographic data which included chronic diseases, age, gender, place of residency, marital status, educational level, employment, and monthly income.

As well as the clinical factors, in the CKD screening index tool tests, participants were asked about their knowledge regarding kidney function, risk factors or causes of kidney disease, kidney disease symptoms, and management of kidney disease as well as the current practice and attitude of patients towards preventing kidney disease, such as eating a well-balanced diet, maintaining regular exercise, following certain restrictions, seeking medical help if needed. It was conducted from December 2021 to May 2022 by a self-administered questionnaire.

The sampling of this study was non-probability convenience sampling. According to the sample size calculation, the sample size was 385 or more. All adult (age ≥ 18 years) hypertensive, and/or diabetic patients, with a family history of CKD, having chronicity with analgesia, or aged more than 65 years were included in the study. However, any patient younger than 18 years old was excluded.

Continuous data were checked for normality and a comparison of means was done using parametric or non-parametric tests. The Statistical Package for the Social Sciences (SPSS) Software was utilized to analyze the data. The data were presented as frequency and in tables; chi-square test, which is a feature of SPSS under cross table or nonparametric analysis, was used to attain a p-value between categorical data-dependent and independent to estimate the association where p-value ≤ 0.05 is considered significant. The data were kept confidential and only used for the purposes described in the study objectives.

This study was approved by the institutional review board (IRB) of Imam Mohammad Ibn Saud Islamic University with approval number (181-2021). Additionally, this research was carried out in accordance with the Code of Ethics of the World Medical Association (Declaration of Helsinki 2013) for experiments involving humans. All participants were aware of the study aims, and written electronic consent was taken before answering the electronic questionnaire. All the data collected from this study were highly confidential and only used for the purposes described in the study objectives. Any identifying information such as names, phone, or fax numbers, medical record number, initials, anywhere in the paper were concealed. No personal information was taken, to ensure privacy.

## 3. Results

Two hundred and ninety-three participants (62.8% males, 37.2% females) took part in this study, as shown in Table 1. From this group, about 44.4% were found to suffer from hypertension, 30.0% from diabetes mellitus, and 25.6% from hypertension and diabetes mellitus. In terms of age, about 23.9% were aged above 60 years old, 72.7% were aged between 30 to 60 years old, and 3.1% were aged between 18 to 30 years old. Residence-wise, 73.4% were living in the Central Region, 19.5% were living in the Western Region, 4.4% were living in the Southern Region, 1.4% were living in Eastern Region, and 1.4% were living in the Northern Region. In terms of marital status, about 90.8% were married, 4.4% were divorced, 2.4% were widowed, and 2.4% were single. For educational level, about 37.2% earned their bachelor’s degree, 22.5% earned a Doctor of Philosophy degree, 22.5% obtained a Master’s degree, 22.5% graduated from high school, and the remaining 9.2% did not finish high school. In terms of employment, about 45.7% were employed, 32.8% were retired, 14.0% were unemployed, 4.4% were unable to work, and 3.1% were students. Monthly income-wise, about 35.2% received above 15,000 Saudi riyals, 22.2% received between 5000 and 10,000 Saudi riyals, 21.5% received below 5000 Saudi riyals, and 21.2% received between 10,000 and 15,000 Saudi riyals.

Each participant was tested in terms of their knowledge about CKD prevention and early detection as shown in Table 2. Twenty-four questions were asked and they were divided into four knowledge question groups: kidney function, risk factors or causes of kidney disease, kidney disease symptoms, and management of kidney disease. Statistical analysis revealed no significant variation of knowledge on CKD prevention and early detection across patient groups who were suffering from hypertension, diabetes mellitus, and both hypertension and diabetes mellitus. Across different age groups, there is no significant difference in the knowledge of patients of CKD prevention and early detection. Similar trends were also observed for other factors such as gender, educational level, and monthly income. On the other hand, a significant difference (*p* = 0.034) in terms of knowledge about kidney function was observed across patients with varying employment status. For other questions on knowledge about CKD prevention and early detection, no statistical difference was observed across patient groups with different employment statuses.

Each participant was also tested in terms of their attitude towards preventing kidney disease as provided in Table 3. Eighteen questions were asked and they were divided into six attitude factor groups. Age, gender, and employment status did not pose a significant difference in the overall attitude of patients towards preventing kidney disease. For patient groups with varying chronic diseases, there was a significant difference in terms of religious attitude and bodily authority (*p* = 0.045). Monthly income (*p* = 0.003) plays a significant factor in the patient attitude about healthcare towards preventing kidney disease. On the other hand, educational level (*p* = 0.012) significantly affects the overall attitude of patients towards preventing kidney disease. Each participant was also asked questions on their current practices to prevent kidney disease as exemplified in Table 4. Overall, chronic disease, age, gender, educational level, employment status, and monthly income did not pose a significant difference on the current practices of patients to prevent kidney disease.

## 4. Discussion

CKD is a serious worldwide health problem with a rising incidence and prevalence [6]. Patients with chronic diseases such as diabetes, hypertension, and cardiovascular diseases are known as high-risk groups for developing CKD. Worldwide, CKDs are the twelfth leading cause of death and the seventeenth leading cause of disability, respectively. About 10 to 13% of the general population had one of the CKDs, counting more than 500 million persons worldwide. Early diagnosis and treatment of CKD will play an important role in delaying CKD progression [7].

Based on the results, knowledge about kidney function had a significant difference across patient groups with varying employment status. Monthly income is a significant factor in patient attitude about healthcare towards preventing kidney disease. On the other hand, educational level significantly affects the overall attitude of patients towards preventing kidney disease. In a study conducted by Yusoff and colleagues in Malaysia, the majority of respondents had poor knowledge, but most of them had a good attitude and good practices towards the risk of CKD. Monthly income, occupation, and educational attainment were found to be significantly correlated with knowledge about CKD [8]. In Palestine, age and educational attainment were deemed significantly associated with higher CKD knowledge scores [5]. In terms of knowledge about CKD, a study in Saudi Arabia revealed that 11.3% of the participants were found to believe that CKD has no specific symptoms [9]. In Jordan, a study revealed that most of the participants have knowledge about kidney disease. However, about half had the wrong information on CKD signs and symptoms [4]. In Tanzania, living in an urban area and educational attainment were strongly linked with a high CKD knowledge score [10]. Another study showed that late-stage CKD and long-time CKD diagnosis can be correlated with a higher knowledge score [11]. Alobaidi [3] explored the knowledge of CKD among the Saudi population and found that disease knowledge was correlated with age, educational attainment, monthly income, civil status, habit for physical activity, and history of medical disease.

In Saudi Arabia, it has been reported that there was a relatively high positive attitude toward CKD prevention and control among the educated Saudi population [12]. Based on the study results, educational level significantly affects the overall attitude of patients towards preventing kidney disease. Yusoff and co-workers mentioned that occupation, marital status, sex, and age were deemed to be significant predictors of a good attitude towards CKD [8]. Additionally, age, monthly income, and high knowledge score were the factors substantially linked with higher CKD attitude scores [4]. Attitudes were also found to be characterized by frequent concern about the health, economic, and social impact of kidney disease [10]. In Indonesia, a positive attitude was reported among pre-dialysis patients with higher scores reported in terms of hope for recovery and diet [13].

Khalil and Abdalrahim [4] had found out that a disparity in knowledge and positive attitude are substantially influenced by income and educational attainment. This result was found to be congruent with the study of Wolf and colleagues [14], which suggests that socio-economic factors including income, educational attainment, and professional standing are independently correlated with knowledge and attitude among diabetic patients with renal dysfunction.

Based on the study results, chronic disease, age, gender, educational level, employment status, and monthly income did not pose a significant difference for the current practices of patients to prevent kidney disease. On the other hand, patients with higher knowledge and attitudes scores, a male gender, and normal body mass index, were statistically significantly associated with a higher practice score towards CKD prevention [5]. Furthermore, healthy practices were noted as being associated with old age as high-risk elderly patients tend to stick to their dietary restrictions [15].

## 5. Conclusions

Socio-demographic factors can play a pivotal role for the understanding of high-risk patients towards prevention and early detection of chronic kidney disease. Understanding knowledge, attitudes, and practices associated with CKD is vital to informing optimal policy and public health responses in the country.

## Figures and Tables

**Table 1 ijerph-20-00871-t001:** Socio-demographic characteristics of the study participants.

Demographics		Count	%
Total		293	100.0
Do you suffer from any chronic disease?	Hypertension	130	44.4
	Diabetes mellitus	88	30.0
	Hypertension and Diabetes mellitus	75	25.6
Age	<18	1	.3
	18–30	9	3.1
	30–60	213	72.7
	>60	70	23.9
Gender	Male	184	62.8
	Female	109	37.2
Region of residence	Eastern Region	4	1.4
	Western Region	57	19.5
	Central Region	215	73.4
	Northern Region	4	1.4
	Southern Region	13	4.4
Marital status	Married	266	90.8
	Single	7	2.4
	Divorced	13	4.4
	Widow	7	2.4
Educational Level	Below high school	27	9.2
	High school	66	22.5
	Bachelor	109	37.2
	Master	25	8.5
	PhD	66	22.5
Employment	Employee	134	45.7
	Unemployed	41	14.0
	Retired	96	32.8
	Student	9	3.1
	Unable to work	13	4.4
Monthly income	Below 5000	63	21.5
	5000–10000	65	22.2
	10,000–15,000	62	21.2
	Above 15,000	103	35.2

**Table 2 ijerph-20-00871-t002:** Statistical analysis on participants’ knowledge about CKD prevention and early detection.

Demographics		Total	Kidney Function	Risk Factors/Causes of Kidney Disease	Kidney Disease Symptoms	Management of Kidney Disease	Knowledge about Chronic Kidney Disease Prevention and Early Detection Score
Do you suffer from any chronic disease?	Hypertension	130	2.06 ± 0.9	3.55 ± 2.0	5.38 ± 2.8	2.00 ± 1.4	13.00 ± 5.5
	Diabetes mellitus	88	1.94 ± 0.9	3.50 ± 1.9	5.52 ± 2.7	1.92 ± 1.3	12.89 ± 5.2
	Hypertension and Diabetes mellitus	75	2.09 ± 0.9	3.87 ± 2.0	5.87 ± 2.9	2.05 ± 1.3	13.88 ± 5.8
*p*-value			0.505	0.446	0.495	0.811	0.448
Age	<30	10	1.90 ± 0.7	3.70 ± 1.9	4.80 ± 2.8	2.20 ± 1.3	12.60 ± 5.9
	30–60	213	2.06 ± 0.9	3.65 ± 2.0	5.76 ± 2.8	2.01 ± 1.3	13.48 ± 5.4
	>60	70	1.97 ± 0.9	3.51 ± 2.0	5.03 ± 2.8	1.89 ± 1.3	12.40 ± 5.7
*p*-value			0.683	0.880	0.118	0.687	0.341
Gender	Male	184	1.98 ± 0.9	3.66 ± 2.0	5.30 ± 2.9	1.98 ± 1.4	12.92 ± 5.6
	Female	109	2.12 ± 0.8	3.55 ± 2.0	5.96 ± 2.7	2.01 ± 1.2	13.64 ± 5.2
*p*-value			0.188	0.656	0.052	0.842	0.279
Educational Level	Below high school	27	1.78 ± 0.9	3.15 ± 2.2	6.33 ± 2.4	1.70 ± 1.3	12.96 ± 4.9
	High school	66	2.05 ± 0.9	3.44 ± 1.8	5.36 ± 2.6	2.00 ± 1.3	12.85 ± 5.2
	Bachelor’s	109	2.03 ± 0.9	3.54 ± 2.0	5.42 ± 2.9	1.99 ± 1.3	12.98 ± 5.6
	Master’s	25	1.80 ± 1.0	3.52 ± 1.8	4.56 ± 2.6	1.68 ± 1.3	11.56 ± 4.5
	PhD	66	2.23 ± 0.8	4.15 ± 2.1	6.00 ± 2.9	2.21 ± 1.4	14.59 ± 6.0
*p*-value			0.131	0.133	0.114	0.346	0.137
Employment	Employee	134	2.18 ± 0.9 ^A^	3.81 ± 2.0	5.81 ± 2.9	2.06 ± 1.4	13.86 ± 5.6
	Unemployed/Unable to work/Student	63	1.87 ± 0.8 ^B^	3.46 ± 1.9	5.54 ± 2.5	1.92 ± 1.3	12.79 ± 5.0
	Retired	96	1.94 ± 0.9 ^B^	3.46 ± 2.0	5.19 ± 2.8	1.94 ± 1.3	12.52 ± 5.6
*p*-value			0.034 ^a,b^	0.331	0.250	0.708	0.154
Monthly income	Below 5000	63	1.94 ± 0.9	3.51 ± 2.0	5.33 ± 2.6	1.84 ± 1.3	12.62 ± 5.2
	5000–10,000	65	2.11 ± 0.9	3.26 ± 2.0	6.12 ± 2.8	2.15 ± 1.4	13.65 ± 5.4
	10,000–15,000	62	2.10 ± 0.9	3.55 ± 1.9	5.39 ± 2.8	1.66 ± 1.0	12.69 ± 5.3
	Above 15,000	103	2.01 ± 0.8	3.95 ± 1.9	5.42 ± 3.0	2.17 ± 1.4	13.55 ± 5.8
*p*-value			0.665	0.153	0.319	0.054	0.556

^a^-significant using One-Way ANOVA Test at <0.05 level. ^b^-Post-hoc Test = LSD. CAPITAL letters indicate Post-Hoc multiple pairing summary indicator. Having the same letter means the same measure statistically.

**Table 3 ijerph-20-00871-t003:** Statistical analysis on participants’ attitudes toward preventing kidney disease.

Demographics		Total	Factor 1	Factor 2	Factor 3	Factor 4	Factor 5	Factor 6	Attitudes toward Preventing Kidney Disease Score
Do you suffer from any chronic disease?	Hypertension	130	17.46 ± 2.7	6.90 ± 2.6 ^A^	5.48 ± 1.6	7.45 ± 2.3	4.47 ± 1.4	5.83 ± 1.7	47.58 ± 6.1
	Diabetes mellitus	88	17.67 ± 3.0	7.67 ± 2.5 ^B^	5.80 ± 1.6	6.84 ± 1.8	4.28 ± 1.4	5.81 ± 1.6	48.07 ± 5.2
	Hypertension and Diabetes mellitus	75	17.33 ± 2.4	6.79 ± 2.6 ^A^	5.57 ± 1.7	7.03 ± 2.5	4.64 ± 1.2	5.69 ± 1.9	47.05 ± 5.6
*p*-value			0.725	0.045 ^a,b^	0.350	0.118	0.241	0.850	0.531
Age	<30	10	17.40 ± 2.1	9.00 ± 3.3	6.10 ± 1.4	6.80 ± 2.1	3.90 ± 1.1	5.10 ± 2.0	48.30 ± 7.2
	30–60	213	17.61 ± 2.8	7.07 ± 2.5	5.62 ± 1.7	7.23 ± 2.3	4.47 ± 1.4	5.88 ± 1.7	47.86 ± 5.5
	>60	70	17.16 ± 2.6	6.93 ± 2.6	5.47 ± 1.4	7.00 ± 1.8	4.50 ± 1.3	5.61 ± 1.6	46.67 ± 6.1
*p*-value			0.489	0.056	0.486	0.667	0.407	0.227	0.296
Gender	Male	184	17.63 ± 2.3	7.03 ± 2.7	5.60 ± 1.6	7.01 ± 2.3	4.57 ± 1.3	5.72 ± 1.8	47.57 ± 5.9
	Female	109	17.26 ± 3.3	7.22 ± 2.4	5.59 ± 1.6	7.41 ± 2.1	4.27 ± 1.4	5.90 ± 1.6	47.64 ± 5.4
*p*-value			0.258	0.548	0.934	0.128	0.061	0.392	0.912
Educational Level	Below high school	27	17.11 ± 2.4	6.44 ± 2.4	5.56 ± 1.2	7.00 ± 2.1	4.48 ± 1.3	5.44 ± 1.7	46.04 ± 5.9 ^A^
	High school	66	17.14 ± 2.9	6.59 ± 2.5	5.21 ± 2.0	7.18 ± 2.3	4.36 ± 1.1	5.65 ± 1.8	46.14 ± 5.0 ^A^
	Bachelor’s	109	17.31 ± 3.0	7.20 ± 2.6	5.78 ± 1.3	7.08 ± 2.1	4.42 ± 1.4	5.84 ± 1.7	47.64 ± 5.8 ^AB^
	Master’s	25	18.20 ± 2.7	7.72 ± 3.1	5.60 ± 1.8	7.00 ± 2.9	4.32 ± 1.8	5.48 ± 1.8	48.32 ± 6.0 ^AB^
	PhD	66	18.03 ± 2.0	7.48 ± 2.4	5.70 ± 1.6	7.38 ± 2.1	4.65 ± 1.3	6.09 ± 1.6	49.33 ± 5.8 ^B^
*p*-value			0.172	0.115	0.238	0.903	0.731	0.336	0.012 ^a,b^
Employment	Employee	134	17.72 ± 2.8	7.32 ± 2.7	5.66 ± 1.6	7.18 ± 2.5	4.43 ± 1.5	5.88 ± 1.7	48.19 ± 5.6
	Unemployed/Unable to work/Student	63	17.08 ± 3.0	7.11 ± 2.5	5.52 ± 1.6	7.35 ± 1.8	4.41 ± 1.2	5.86 ± 1.6	47.33 ± 5.7
	Retired	96	17.45 ± 2.5	6.79 ± 2.5	5.56 ± 1.7	7.00 ± 2.0	4.52 ± 1.3	5.61 ± 1.7	46.94 ± 5.8
*p*-value			0.306	0.309	0.835	0.617	0.850	0.473	0.244
Monthly income	Below 5000	63	17.13 ± 3.0	6.76 ± 2.6	5.52 ± 1.5	7.35 ± 2.3	4.11 ± 1.0 ^A^	5.56 ± 1.8	46.43 ± 5.9
	5000–10,000	65	17.57 ± 2.2	6.82 ± 2.4	5.48 ± 1.8	7.42 ± 2.1	4.31 ± 1.4 ^AB^	5.63 ± 1.8	47.22 ± 4.6
	10,000–15,000	62	17.74 ± 3.0	7.48 ± 2.6	6.02 ± 1.1	6.97 ± 2.1	4.34 ± 1.5 ^AB^	6.00 ± 1.5	48.55 ± 6.1
	Above 15,000	103	17.51 ± 2.7	7.26 ± 2.6	5.47 ± 1.7	6.99 ± 2.3	4.83 ± 1.4 ^B^	5.90 ± 1.7	47.97 ± 6.0
*p*-value			0.637	0.302	0.141	0.496	0.003 ^a,c^	0.367	0.168

^a^-significant using One-Way ANOVA Test at <0.05 level. ^b^-Post-hoc Test = LSD. ^c^-Post-hoc Test = Games-Howell. CAPITAL letters indicate Post-Hoc multiple pairing summary indicator. Having the same letter means the same measure statistically.

**Table 4 ijerph-20-00871-t004:** Statistical analysis on participants’ current practice to prevent kidney disease.

Demographics		Total	Current Practice to Prevent Kidney Disease Score
Do you suffer from any chronic disease?	Hypertension	130	35.75 ± 5.3
	Diabetes mellitus	88	34.74 ± 5.9
	Hypertension and Diabetes mellitus	75	36.11 ± 6.2
*p*-value			0.265
Age	<30	10	35.40 ± 6.5
	30–60	213	35.57 ± 6.1
	>60	70	35.47 ± 4.3
*p*-value			0.989
Gender	Male	184	35.16 ± 5.2
	Female	109	36.17 ± 6.4
*p*-value			0.142
Educational Level	Below high school	27	37.15 ± 5.6
	High school	66	36.12 ± 6.3
	Bachelor’s	109	34.97 ± 5.7
	Master’s	25	35.60 ± 5.7
	PhD	66	35.21 ± 5.0
*p*-value			0.389
Employment	Employee	134	34.76 ± 5.9
	Unemployed/Unable to work/Student	63	36.63 ± 5.2
	Retired	96	35.91 ± 5.6
*p*-value			0.073
Monthly income	Below 5000	63	35.89 ± 6.2
	5000–10,000	65	35.62 ± 5.6
	10,000–15,000	62	34.97 ± 5.9
	Above 15,000	103	35.62 ± 5.4
*p*-value			0.829

## Data Availability

https://docs.google.com/file/d/1oAm6kA9ZEQsdCg52KPRVpz6lo5N4yaGl/edit?usp=docslist_api&filetype=msexcel (accessed on 1 March 2022).

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
