# Peer review of "Knowledge, Attitudes, and Practices of High-Risk Patients towards Prevention and Early Detection of Chronic Kidney Disease (CKD) in Saudi Arabia"

_ijerph, 2023, doi:10.3390/ijerph20010871_

Round 1
Reviewer 1 Report
Thank you for the opportunity to offer a peer review - please see attached file for comments/suggestions.

Reviewer 2 Report
Major revision is required.
Methods
1. The sample size (n=293) of this study was lower than the minimal sample size (n=385) when non-probability convenience sampling was used. Was the power of this study adequate?
2. The description was not clear in the method section. Where do you get a sample from in research? How are data collection instruments prepared? What is CKD screening index tool? The statistical description was also not clear. How to perform statistical analysis? Did you do multiple linear regression analysis to determine the association between clinical and socio-demographic factors and practices?
Author Response
1- Sample was taken through online survey distributed via social media and database of patients have (CKD,HTN,DM) from medical center of imam university in Riyadh- KSA
2- Survey was taken from other research named Development and psychometric evaluation of the Chronic Kidney Disease Screening Index, which was used in our research and cited in our research paper references after taking the permission from the authors. Also we did pilot study before using the instrument.
- what is CKD screening index tool? Newly developed valid and reliable instrument developed by the authors of the research paper named Knowledge, attitudes, and practices towards prevention and early detection of chronic kidney disease. Which is cited in the references.
-statical analysis was performed through SPSS ? Chi square test is a feature of SPSS under cross table or nonparametric analysis
3- Please see the attachment

Round 2
Reviewer 1 Report
Please see file attached.

Author Response
Introduction:
- This might be re-phrased to something such as, However, there is limited information on the knowledge, attitudes, and practices towards prevention and early detection of CKD in Saudi Arabia. Thank you for your comment and note, we re-phrased it in the manuscript.
- Further background information such as health initiatives which do/don t alert the public to risks (how would a patient know?) that’s what we were trying to understand in our study, the factor that affect the patient awareness towards the risk.
- which health professional(s) would screen for health behaviours as opposed to clinical factors that place patients at risk? We are not sure if we understand the question clearly but patients at risk of CKD should have regular follow up appointment either with their family physician or their primary physician depends on the risk they have (DM-HTN- analgesia use- history of kidney disease).
Methodology:
- It would be helpful to add whether the questionnaire/screening index and sociodemographic items were customised in any way for the Kingdom as health systems, diet or other influences might vary across the middle East. Actually the questionnaire was developed in the middle east (Jordan), so it is customized for the middle east, and no further items customization were needed for the kingdom as a health system.
- whether any items were re-coded for data analysis. No.
- HOW participants were aware of the study aims. It was written and explained clearly in Arabic and English languages, in the first page with the consent form, so when they read the aim of the study they precede and sign the consent form if they accept it.
- whether they used a research device or one of their own since the questionnaire was stated to be electronic. As we mentioned the electronic questionnaire was distributed through social media, so they had to use their own devices in any time they want.
- Issues of concern would be confidentiality and privacy if for example participants were completing the questionnaire in an open space such as a waiting room or in a room with other people. They could complete the survey in any suitable time for them and when they started the questionnaire their names were not written, so if someone is next to them and can see their device, he would not know this information is for how? Because it is collected without names.
- it would also be helpful if the participant information sheet for the questionnaire was addressed in more detail as health literacy and understanding, diet, gender roles and many other variables may vary across patient groups. It is already addressed in line 47 ( socio-demographic data which included chronic diseases, age, gender, place of residency, Marital status,educational level, employment and monthly income).
- Line 47 in Methods - No personal information was taken to ensure their privacy which sorts of information was NOT collected? This statement was mentioned in line 67 not 47 and the information that was not collected were mentioned already also (Any identifying information such as names, phone or fax numbers, medical record number, initials, anywhere in the paper were concealed. No personal information was taken to insure their privacy).
- Clarity around what was/was not collected is anticipated. Mentioned in line 66 and 67 (Any identifying information such as names, phone or fax numbers, medical record number, initials, anywhere in the paper were concealed. No personal information was taken to insure their privacy. The aim was to have the participant information for the research but without knowing his or her identity to maintain their privacy.
- Storage security of the data similarly. Mentioned in line 65, the data was collected electronically using google survey tool and kept electronic and confidential for the statistical analysis.
Results:
- Table 3 reports demographics by Factors which will benefit from being defined/discussed as an attitudes score seems to result from a summation across the factors. Which factors were identified in this study? The demographic factors that were identified in Table 3 were age, gender, employment status, presence of chronic diseases, monthly income, and educational level. How do the factors vary? These factors vary in a sense that each factor aimed to answer the disparities in terms of knowledge, attitudes, and practices among high-risk patients toward prevention and early detection of chronic kidney disease. Socio-demographic factors can play a pivotal role towards understanding of high-risk patients towards prevention and early detection of chronic kidney disease. Understanding knowledge, attitudes, and practices associated with CKD is vital to informing optimal policy and public health responses in the country. Because the majority of CKD patients are not clinically identified, this global growth in the prevalence of CKD necessitates improvements in the global approach to CKD prevention, primarily via recognizing risk factors, because the majority of CKD cases were not clinically recognized mainly because of the lack of patients’ awareness about CKD risk factors. The most important risk factors for developing CKD are hypertension and diabetes mellitus. These two diseases promote vascular alterations that increase the risk for macro and micro vascular complications including kidney impairment. What domain does each address? Khalil and Abdalrahim [4] had found out that disparity in knowledge and positive attitude are substantially influenced by income and educational attainment. This result was found out to be congruent with the study of Wolf and colleagues [14] as to which suggests that socio-economic factors including income, educational attainment, and professional standing are independently correlated with knowledge and attitude among diabetic patients with renal dysfunction. This study focused on assessing the knowledge, attitudes, and practices of high-risk patients towards prevention and early detection of chronic kidney disease in Saudi Arabia as well as finding out the association between the participants' knowledge and attitude with their socio-demographic characteristics. Employment status played a substantial impact on knowledge towards kidney function, monthly income plays a significant factor on patient attitude on healthcare towards preventing kidney disease, and educational level significantly affects the overall attitude of patients towards preventing kidney disease. Patient practices on kidney disease prevention were not substantially affected by chronic disease, age, gender, educational level, employment status, and monthly income.
- Clarity may be enhanced by further details such as how were the six attitude factor groups developed in this study was it by Factor analysis? If so was it PAF? Varimax rotation? Which features of SPSS were used? chi-square test which is a feature of SPSS under cross table or nonparametric analysis was used to attain a p-value between categorical data-dependent and independent to estimate the association where p-value ≤ 0.05 is considered significant.
- Description of Table 4 was also brief, and clarity could be enhanced ach participant was also asked questions on their current practices to prevent kidney disease as exemplified in Table 4.Which were the items/questions? It was 11 questions which were , eat well balanced meals, exercise regularly, have regular check-ups, keep normal wight, do not smoking, do not drink alcohol, follow my medications regimen, follow my food restriction, recognise abnormal changes related to kidney disease, seek medical help if I notice sign od kidney disease, get family help and support if I get kidney disease.

Reviewer 2 Report
This revision can be accepted for publication.
Author Response
Thank you very much for your comments.